# A rapid review to identify physical activity accrued while playing golf

Jack Luscombe,[1,2] Andrew D Murray,[2,3] Evan Jenkins,[1,2] Daryll Archibald[4,5]

[1]Medical School, University of Edinburgh, Edinburgh, UK
[2]Physical Activity for Health Research Centre, University of Edinburgh, Edinburgh, UK
[3]Department of Sport and Exercise, University of Edinburgh, Edinburgh, UK
[4]Scottish Collaboration for Public Health Research and Policy, University of Edinburgh, Edinburgh, UK
[5]School of Psychology and Public Health, La Trobe University, Melbourne, Australia

**Correspondence to**
Jack Luscombe;
s1204551@sms.ed.ac.uk

## ABSTRACT

**Objective** To identify physical activity (PA) accrued while playing golf and modifiers of PA accrued.

**Design** A rapid review of primary research studies. Quality was assessed using the National Heart, Lung, and Blood Institute quality assessment tool for cohort and cross-sectional studies.

**Methods and outcomes** The following databases were searched from 1900 to March 2017: SPORTDiscus, Web of Science, PsycINFO, MEDLINE, Google Scholar, Google Advanced Search, ProQuest, WHO International Clinical Trials Registry Platform. All primary research investigating golf or golfers with any of the following outcomes was included: metabolic equivalent of task, oxygen uptake, energy expenditure, heart rate, step count, distance covered, strength, flexibility, balance, sedentary behaviour.

**Results** Phase one searching identified 4944 citations and phase two searching identified 170 citations. In total, 19 articles met inclusion criteria. Golf is primarily a moderate intensity PA, but may be low intensity depending on the playing population and various modifiers. Less PA is accrued by those who ride a golf cart compared with those walking the course.

**Conclusions** Golf can be encouraged in order to attain PA recommendations. Further research is required into the relationship between golf and strength and flexibility PA recommendations and how modifiers affect PA accrued.

**PROSPERO registration number** CRD42017058237.

## INTRODUCTION

Moderate intensity physical activity (PA) is known to provide longevity, physical and mental health benefits.[1–4] PA guidelines[1] generally recommend, for adults, at least 150 min of moderate intensity activity or 75 min of vigorous PA per week or a combination of the two. In addition, PA to improve muscle strength on at least 2 days a week and efforts to minimise the amount of time spent sedentary are recommended. An estimated 41%–51% of women and 32%–41% of men do not meet these guidelines[5 6] in the UK. Furthermore, the proportion of adults meeting guidelines decreases with age—only 7%–36% of adults aged 75 and over meet the recommendations.[5 6]

Golf is a popular sport played by over 50 million people[7] of all ages and abilities in over 200 countries.[8] In contrast to most sports, participation is higher in middle-aged and older adults.[9–11] Reviews and guideline documents have suggested golf can provide moderate intensity[1 12–14] and muscle-strengthening PA.[13] These studies have not formally assessed the quality of the evidence.

The frequently cited Compendium of Physical Activities[11] is a classification of intensity costs of various physical activities. It lists golf as, on average, providing 4.8 metabolic equivalents of task of PA, a moderate intensity.

A recently published systematically conducted scoping review[10 15] provided an overview of golf and health and further highlighted that golf can provide moderate intensity PA. As per standard guidelines for undertaking scoping reviews,[16] the relative strengths and limitations of included studies were not assessed. There have been no other reviews found that use systematic methods exploring PA and golf. We therefore aimed to provide a rapid review to identify PA accrued while playing golf.

Murray *et al's*[10] scoping review noted several factors that influence the intensity of PA while playing golf: use of a golf-cart, course profile, age, weight, sex and baseline fitness of participants.[10] Our secondary aim was therefore to report modifiers to the amount of PA accrued while playing golf.

## METHODS

Our systematic review adhered to our published protocol[17] and followed Preferred

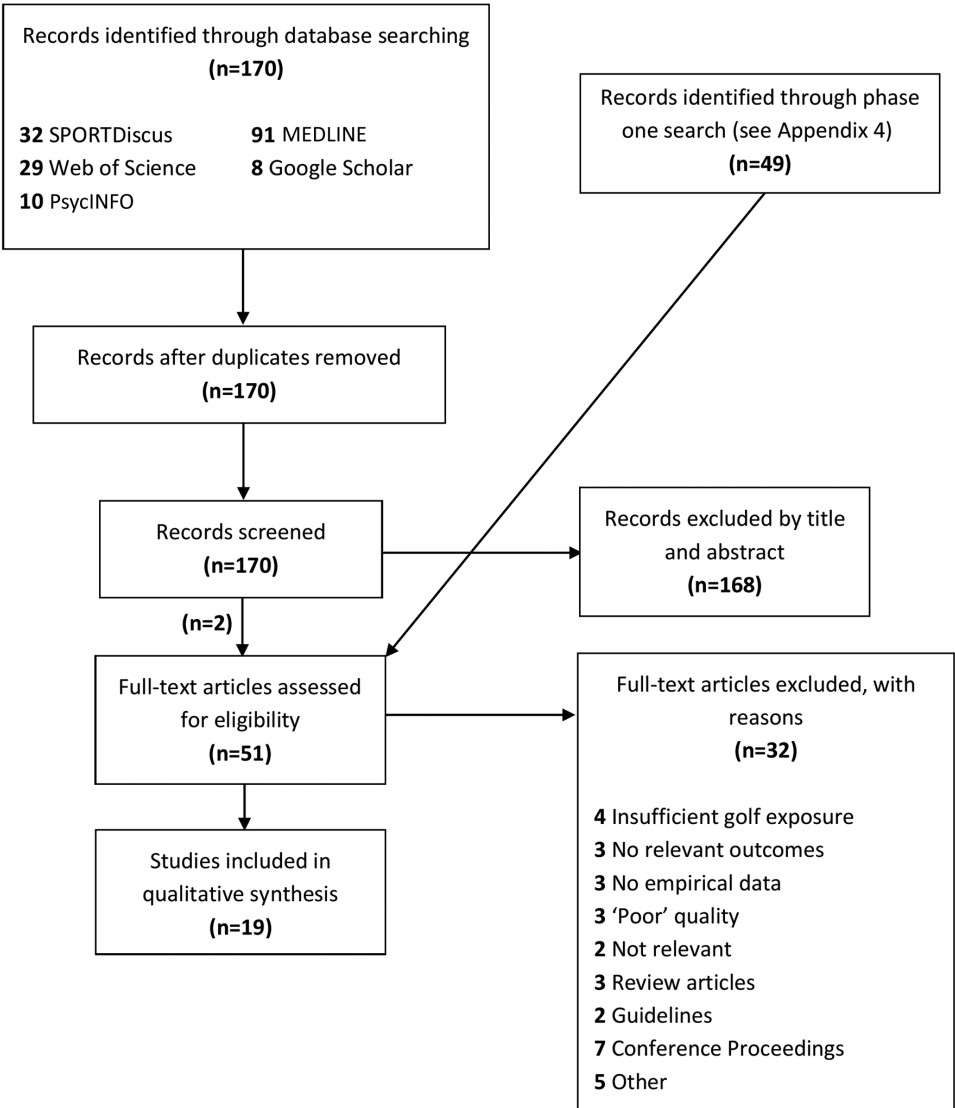

**Figure 1** Results of systematic electronic search.

Reporting Items for Systematic Reviews and Meta-Analyses (PRISMA) guidelines.[18]

Rapid reviews are a streamlined, time efficient and pragmatic approach to synthesise evidence. They have been shown to produce similar conclusions to systematic reviews.[19] Variable methodologies have been described[20] and therefore guidance was sought as to the best approach.[21] A rapid review was conducted due to a short time-frame in which to complete the research. To streamline the process, the search strategy from a recent scoping review[10] was used and adapted, there were less exhaustive searches of grey literature and only one reviewer assessed all papers for selection, data extraction and quality assessment compared with a full systematic review. Similar concessions have been described in the literature.[22]

### Search strategy

We adopted a two-phase search strategy. The first phase employed the search strategy used in the recently published scoping review published by team members[10]—a precursor to this rapid review. The scoping review search was undertaken in November 2015 across the following databases: SPORTDiscus, Web of Science, PsycINFO, MEDLINE, Google Scholar, Google Advanced Search, ProQuest dissertations, WHO International Clinical Trials Registry Platform. The search identified 301 studies relating to the scoping review's aims—the relationship and effects of golf on physical and mental health. Forty-nine of those studies were found to be specifically related to golf and PA, which will be used in the current review.

The second phase of the search strategy involved adapting and updating the scoping review search. The search was rerun restricting its scope to search for papers related to golf and PA only that were published from November 2015 to March 2017. A focused grey literature search was performed using the modified terms 'golf AND health'. The full search strategies can be found in online supplementary appendices 1 and 2.

## Study selection

One reviewer (JL) selected studies for review first by title and abstract, then by full text review, against inclusion/exclusion criteria with the exception of title and abstract screening of phase two results, conducted by DA. A second reviewer (EJ) independently reviewed a random sample of 10% of the papers by full text review for inclusion or exclusion. Concordance was checked and any discrepancies were discussed and resolved by a team member, either ADM or DA.

Inclusion and exclusion criteria were developed through researcher discussion:

Inclusion criteria
► Research articles not limited by geographical location, language or setting.
► Research articles published since 1900 up until March 2017.
► Research articles discussing any of the following outcomes in relation to golf: metabolic equivalent of task (MET), oxygen uptake, energy expenditure (EE), heart rate (HR), step count, distance covered, strength, flexibility, balance, sedentary behaviour.
► Any form of playing golf (including but not limited to 18 holes, 9 holes, driving range) or research involving golfers.
► All ages groups and both sexes of participants.
► Sources of information including randomised control trials, cohort, case-control and cross-sectional studies, that have been synthesised quantitatively.

Exclusion criteria
► Studies focussing exclusively on caddies and/or spectators.
► Qualitative studies, reviews, opinion pieces, magazine and newspaper articles, case reports, conference proceedings.

## Data extraction

Data were extracted by one reviewer (JL) using a data extraction form. The data extraction form was piloted using 10% of papers and modifications were made. A further random sample of 10% was independently extracted by a second reviewer (EJ) and results compared. Concordance was checked and any discrepancies were discussed and resolved by a team member, either ADM or DA. A sample data extraction form can be found in online supplementary appendix 3.

## Quality assessment

Our protocol[17] details use of the Effective Public Health Practice Project's quality assessment tool for quantitative studies[23] to assess study quality. After trialling, it became apparent the tool was more suited to interventional studies with groups. As the large majority of included studies are observation cross-sectional design, the tool was not suitable and therefore the National Heart, Lung, and Blood Institute quality assessment tool for observational cohort and cross-sectional studies[24] was used. Eligible studies were assessed by one reviewer (JL). A second reviewer (EJ) independently assessed a random sample of 10% of the papers using the same tool. Concordance was checked and any discrepancies were discussed and resolved by a third researcher, either ADM or DA. Studies rated 'Fair' or 'Good' were included in the review.

## Data synthesis and analysis

Due to the wide heterogeneity of included studies in terms of study design, population, setting, outcomes and study quality, data were synthesised narratively with summary tables and figures using the following outcomes: MET, EE, oxygen uptake, HR, steps taken, distance covered, strength, flexibility, balance and sedentary behaviour. Modifiers to PA accrued were noted during data extraction and were also narratively synthesised. There were no principal summary measures due to the studies' heterogeneity; data were presented using the raw outcome measures.

## RESULTS

### Study selection

In total, 3550 independent records were identified through our systematic two-phase electronic search. Three thousand three hundred and eighty independent records were identified in phase one.[10] Three thousand and fifteen records were excluded on screening of title and abstract, and 64 articles were excluded by full text review. Of the remaining 301 articles, 49 were specifically related to golf and PA. A flowchart detailing the results of phase one of the search can be found in online supplementary appendix 4.

Phase two of the search identified 170 further independent records (figure 1). One hundred and sixty-eight records were excluded by title and abstract. The 49 articles from phase one were included here and assessed for eligibility by full text review. Thirty-two articles were excluded by full text review. Nineteen articles remained that met the inclusion criteria and were included in the review. Citations of included studies can be found in online supplementary appendix 5.

### Study characteristics

Of the 19 included studies, 11 were conducted in USA, 3 in China and 5 in other countries (Germany, Sweden, Austria and Norway). All included studies were cross-sectional design. Sixteen of the identified studies were primary published research papers, and three were published dissertations. The studies' publication dates ranged from 1970 to 2015. Four of the studies were published pre-2000, and 15 studies were published post-2000.

Ten different outcome measures were used in the review. The most frequently reported were: HR (10 studies), EE (9 studies) and METs (6 studies). No studies reported on sedentary behaviour. Further characteristics of included studies are presented in online supplementary appendix 6.

## Quality of included studies

Information on quality assessment of included studies can be found in online supplementary appendix 7. All studies provided a clear objective or research question. Most studies (74%) did not provide a sample size justification, power description or variance/effect estimates. It was unclear in the majority of studies whether outcome assessors were blinded to exposure status of participants (68%). Five studies were rated 'Good' and 14 were rated 'Fair'.

## Outcomes

### Energy expenditure

Nine studies identified EE as an outcome.[25–33] Eight studies were rated 'Fair'[25–31 33] and one rated 'Good'.[32] Results are detailed in table 1. Two studies found significantly higher EE on hillier courses compared with flatter courses. Zunzer et al,[32] however, found no significant difference in EE between hilly and flat courses.

Lampley et al[30] noted a significantly higher rate of EE in women. In contrast, two studies[32 33] found males expended significantly more energy than females. However, Zunzer et al[32] noted that this is not significant if body mass is accounted for and Tangen et al[33] suggested that this may be due to differences in course distance.

Two studies[25 33] found no significant difference in EE in relation to skill level, despite less skilled players taking a larger number of shots in total and on average being less able to advance the ball accurately.

Crowell[26] noted the lowest EE when riding a golf cart, then pulling clubs and highest when carrying clubs. Zunzer et al[32] found that those who rode a golf cart had significantly lower EE than those who pulled or carried clubs. Tangen et al[33] found no significant difference in relation to club transportation; however, it is noted that this may be due to small sample size in each group.

### Metabolic equivalent of task

Six studies identified MET as an outcome.[27–29 32–34] Four of the studies were rated 'Fair'[27–29 33] and two rated 'Good'.[32 34] Results are detailed in table 2. Dobrosielski et al[28] found a significant difference between patients with cardiac disease and healthy adults in average MET ($57\pm2.7$; $46\%\pm2.6\%$ peak MET) and peak MET ($89\pm3.3$; $77\%\pm3.6\%$ peak MET). However, Unverdorben et al[34] found the same MET value (3.1) for patients with cardiac disease and healthy adults. Zunzer et al[32] noted no significant difference in METs between sexes, whereas Tangen et al[33] found an almost significant difference between men and women (P=0.069). Zunzer et al[32] found no significant difference in METs between hilly and flat golf courses.

### Heart rate

Ten studies reported HR as an outcome.[25 26 29 31–37] Eight were rated 'Fair'[25 26 29 31 33 35–37] and two rated 'Good'.[32 34] Mean HR and mean percentage of maximum HR ($\%HR_{max}$) are presented in table 3. In relation to maximum HR, Stauch et al[35] found that most time during a round of golf is spent at $50\%–74\% HR_{max}$. Tangen et al[33] described that 75% of a golf round is played at $<70\% HR_{max}$ and 25% is $>70\%$. Broman et al[35] found that 70% of total time for elderly men is at $>70\% HR_{max}$, whereas, for middle-aged and younger men, most time is spent at $<70\% HR_{max}$. Loy[31] estimated that 75.25 min are $>60\%$ HR reserve.

One paper[25] noted a significant difference in mean HR and a second paper[36] time spent $>40\% HR_{max}$ between hillier and flatter courses. Two papers found no significant difference in mean HR in relation to course profile.[32 33] However, Tangen et al[33] found a significantly higher maximum HR on the hillier course.

Two papers found highest HRs when carrying clubs, then pulling clubs and lowest when riding a golf cart.[26 36] One of these studies[36] found a significant difference in percentage of time spent $>40\% HR_{max}$ between carrying and pulling clubs and riding a golf cart. Similarly, Zunzer et al[32] found that participants who rode a golf cart had significantly lower mean HR than those who carried or pulled their clubs. Stauch et al[37] observed no significant difference in mean or maximum HR in relation to club transportation. However, it is noted that there are significant differences in ages, a possible modifier to PA attained, between groups—this was also observed in another study.[32]

Crowell[26] noted little difference in mean HR in relation to skill level and Burkett et al[25] found no significant difference. In relation to sex, two papers[32 33] observed no significant difference in mean HR and one paper,[32] minimum, maximum HR or mean percentage $HR_{max}$. Broman et al[35] found that older golfers spent significantly more time at higher $\%HR_{max}$ than middle-aged or younger golfers. Tangen et al[33] found that older golfers ($>50$ years) spent less time at high intensity level ($>120$ bpm) than younger golfers ($<50$ years)—but suggested that this may be due to differences in maximum HR. Unverdorben et al[34] observed no significant difference in mean HR between patients with cardiac disease and healthy controls, but noted that the maximum HR of controls was higher and therefore patients with cardiac disease may work harder.

### Oxygen uptake

Four studies listed oxygen uptake as an outcome.[26 27 31 34] Three were rated 'Fair'[26 27 31] and one rated 'Good'.[34] Results are detailed in table 4. Crowell[26] found that riding a golf cart required least oxygen uptake per minute, then pulling clubs and carrying clubs required the most oxygen uptake per minute. The study also noted that golfers of lower handicaps ($\leq10$) required less oxygen per minute when pulling or carrying clubs than golfers with higher handicaps ($\geq11$). Dear et al's[27] value of $9.9\pm1.7$ mL kg$^{-1}$ min$^{-1}$ equates to $34.4\%\pm9.1\%$ oxygen uptake reserve. Unverdorben et al[34] found that patients with cardiac disease had a significantly higher $\%V_{O_2}$max while playing golf compared with healthy controls.

**Table 1** EE for a round of golf

| Study | Quality assessment | No. of holes | Club transportation | Course profile | EE (kcal min⁻¹) | Net EE (kcal) | Gross EE (kcal) | EE (kcal kg hour⁻¹) |
|---|---|---|---|---|---|---|---|---|
| Burkett and von Heijne-Fisher[25] | Fair | 18 | Carrying clubs | Flat<br>Medium<br>Hilly | 7.25±1.75<br>8.15±1.79<br>8.25±1.83 | – | – | – |
| Crowell[26] | Fair | 9 | Riding a golf cart<br>Pulling clubs<br>Carrying clubs | Not reported | 5.2<br>6.8<br>7.5 | – | – | – |
| Dear et al[27] | Fair | 9 | Pulling clubs | Not reported | – | 310.3±83.9 | 511.6±115.5 | – |
| Dobrosielski et al[28] | Fair | 9 | Pulling clubs | Mixed | – | 458 | – | – |
| Gabellieri[29] | Fair | 18 | Carrying clubs | 'Undulating' | – | – | 1202.8±465.2 | – |
| Lampley et al[30] | Fair | 9 | Pulling clubs | Not reported | – | – | – | 4.2±0.6 (male)<br>4.8±0.4 (female) |
| Loy[31] | Fair | 18 | Carrying clubs | Hilly | 6.2±0.6 | – | – | 4.8* |
| Zunzer et al[32] | Good | 9<br><br>18 | Mixed | Mixed | – | – | 520±133 (male)<br>273±66 (female)<br>926±292 (male)<br>556±180 (female) | – |
| Tangen et al[33] | Fair | 18 | Mixed | Hilly | – | – | 2467 (male)<br>1587 (female) | – |

Please refer to online supplementary appendix 6 for characteristics of the above studies.
*Calculated for a 68 kg man.
EE, energy expenditure.

**Table 2**  MET of a round of golf

| Study | Quality assessment | No. of holes | Club transportation | METs (mean±SD) |
|---|---|---|---|---|
| Dear et al[27] | Fair | 9 | Pulling clubs | 2.8±0.5 |
| Dobrosielski et al[28] | Fair | 9 | Pulling clubs | 4.1±0.1 (cardiac disease) |
| Gabellieri[29] | Fair | 18 | Carrying clubs | 8.6±3.1 |
| Unverdorben et al[34] | Good | 18 | Pulling clubs | 3.1 (cardiac disease) |
| | | 18 | Pulling clubs | 3.1 (controls) |
| Zunzer et al[32] | Good | 9 | Mixed | 2.9±0.8 (male) |
| | | | | 2.2±0.6 (female) |
| | | 18 | Mixed | 2.8±0.7 (male) |
| | | | | 2.1±0.7 (female) |
| Tangen et al[33] | Fair | 18 | Mixed | 5.8 (male) |
| | | | | 4.9 (female) |

Please refer to online supplementary appendix 6 for characteristics of the above studies.
MET, metabolic equivalent of task.

## Steps taken

Three articles were found with steps taken as an outcome.[29 33 38] All studies rated 'Fair' in quality assessment. The included studies all involved an 18-hole round of golf. Studies found that 11245±1351,[29] 11948±1781,[38] 16080±1195 (male)[33] and 16667±992 (female)[33] steps were taken during a round of golf. One study[29] found significant negative correlation between number of steps

**Table 3**  Mean HR and percentage of maximum HR during a round of golf

| Study | Quality assessment | No. of holes | Club transportation | Course profile | Mean HR (bpm) | Mean %HR$_{max}$ |
|---|---|---|---|---|---|---|
| Burkett and von Heijne-Fisher[25] | Fair | 18 | Carrying clubs | Flat | 108.20±13.16 (GS) | – |
| | | | | | 110.80±7.26 (AS) | |
| | | | | Medium | 121.80±18.54 (GS) | |
| | | | | | 117.80±13.54 (AS) | |
| | | | | Hilly | 123.80±21.81 (GS) | |
| | | | | | 116.20±14.97 (AS) | |
| Crowell[26] | Fair | 9 | Riding a golf cart | Not reported | 89.1±10.6 | – |
| | | | Pulling clubs | | 103±9.2 | |
| | | | Carrying clubs | | 113.1±8.8 | |
| Gabellieri[29] | Fair | 18 | Carrying clubs | 'Undulating' | 103.5±13.2 | 55.2±7.4 |
| Loy[31] | Fair | 18 | Carrying clubs | Hilly | 124.7±8.6 | |
| Stauch et al[37] | Fair | 18 | Riding a golf cart | Hilly | 111.0±14.0 | – |
| | | | Pulling clubs | | 107.2±11.0 | |
| | | | Carrying clubs | | 118.4±17.0 | |
| Unverdorben et al[34] | Good | 18 | Pulling clubs | Hilly | 105.4±10.6 (patients with cardiac disease) | – |
| | | | | | 100.5±7.3 (controls) | |
| Zunzer et al[32] | Good | 9 | Mixed | Mixed | 101±12 (male) | 59.2±3.1 (male) |
| | | | | | 99±13 (female) | 59.2±8.9 (female) |
| | | 18 | | | 105±14 (male) | 60.9±8.6 (male) |
| | | | | | 103±12 (female) | 61.6±7.7 (female) |
| Tangen et al[33] | Fair | 18 | Mixed | Hilly | 104.1±14.5 (male) | – |
| | | | | | 110.8±16.9 (female) | |

Please refer to online supplementary appendix 6 for characteristics of the above studies.
AS, average skill (score 80–95); GS, good skill (score <80); HR, heart rate.

**Table 4** Oxygen uptake during a round of golf

| Study | Quality assessment | No. of holes | Club transportation | Oxygen uptake ($l\,min^{-1}$) (mean±SD) | Oxygen uptake ($ml\,kg^{-1}\,min^{-1}$) (mean±SD) | %$V_{O_2}$max |
|---|---|---|---|---|---|---|
| Crowell[26] | Fair | 9 | Riding a golf cart | 1.05±0.11 | 8.5 | – |
| | | 9 | Pulling clubs | 1.37±0.03 | 9.1 | |
| | | 9 | Carrying clubs | 1.50±0.11 | 9.7 | |
| Dear et al[27] | Fair | 9 | Pulling clubs | – | 9.9±1.7 | – |
| Loy[31] | Fair | 18 | Carrying clubs | 1.23±0.11 | – | |
| Unverdorben et al[34] | Good | 18 | Pulling clubs | – | – | 76.0±13.1 (patients with cardiac disease) |
| | | 18 | | | | 55.3±9.1 (controls) |

Please refer to online supplementary appendix 6 for characteristics of the above studies.

taken and: weight of the golf bag (P<0.05), EE (P<0.01) and minimum HR (P<0.01) of participants.

## Distance covered
Five studies detailed distance covered as an outcome.[26 27 29 32 33] Four of the studies were rated 'Fair'[26 27 29 33] and one study[33] was rated 'Good'. Results are detailed in table 5. With the exception of Crowell,[26] all studies estimated between 8.7 and 11.25 km walked for an 18-hole course and 4.4 and 5.32 km for a 9-hole course. Distance covered is highly dependent on the individual golf course length. The course in Crowell's study is poorly described, but this may account for the shorter distance. A much shorter distance (3.18 km) is walked riding a golf cart compared with pulling a golf cart or carrying clubs.[26] There is no notable difference in distance walked when pulling a golf cart compared with carrying clubs. Males walked longer distances than females. Zunzer et al[32] noted a significant difference between male and female distance walked over 18 holes. However, in both studies[32 33] and as is usual on golf courses, the men's course is longer than the women's. Tangen et al[33] found that, when course length is accounted for, women (2.13 times the course length) walked significantly longer than men (1.98 times the course length).

## Strength
One study listed strength as an outcome[39] and rated 'Good'. Sell et al[39] found that golfers with a lower handicap (<0) had significantly greater strength over a range of measures when compared with handicaps of 0–9 and 10–20. Tables are not listed for strength, flexibility or balance outcomes due to the heterogeneity of measurements.

## Flexibility
One study listed flexibility as an outcome[39] and rated 'Good'. Sell et al[39] found that golfers with a lower handicap (<0) had significantly greater range of motion in several measures of shoulder, hip, torso flexibility than golfers with higher handicaps (0–9 and 10–20).

## Balance
Five studies listed balance as an outcome.[39–43] Three studies rated 'Good',[39 40 42] and two studies rated 'Fair'.[41 43]

**Table 5** Distance covered in a round of golf

| Study | Quality assessment | No. of holes | Club transportation | Sex | Distance (km, mean±SD) |
|---|---|---|---|---|---|
| Crowell[26] | Fair | 18 | Riding a golf cart | Male | 3.18±0.56* |
| | | | Pull cart | | 7.37±0.71* |
| | | | Carrying clubs | | 6.47±0.84* |
| Dear et al[27] | Fair | 9 | Pull cart | Male | 4.4±3.6 |
| Gabellieri[29] | Fair | 18 | Carrying clubs | Male | 8.7±0.6* |
| Zunzer et al[32] | Good | 18 | Mixed | Male | 10.54±0.94 |
| | | | Mixed | Female | 9.89±0.81 |
| | | 9 | Mixed | Male | 5.32±0.48 |
| | | | Mixed | Female | 5.25±0.76 |
| Tangen et al[33] | Fair | 18 | Mixed | Male | 11.25±0.83 |
| | | | Mixed | Female | 10.00±0.56 |

Please refer to online supplementary appendix 6 for characteristics of the above studies.
*Converted to kilometres.

Three studies focused on older golfers,[40–42] and all papers found elderly golfers had significantly better balance control when compared with controls over a variety of measures. Tsang et al[41] noted that the balance of elderly golfers was comparable to that of young controls (no significant difference).

Sell et al[39] found that golfers with better handicaps (<0) had significantly better single-leg balance than golfers with handicaps 0–9 and 10–20. Schachten et al[43] noted a significant improvement in patients with stroke after participating in a 10-week, 20-session golf putting intervention. However, a significant improvement was also noted in the comparator group and no significant difference was observed between groups.

## DISCUSSION

EE for an 18-hole round of golf appears to achieve the America College of Sports Medicine's (ACSM) recommendation of $1000 \, kcal \, week^{-1}$,[44] and could be separated into two 9-hole rounds. The length of time a round of golf takes can compensate for the low EE per minute. Fifty per cent of MET values stated are within the range of moderate intensity (3–5.9).[44] Values for $\%HR_{max}$ are within light intensity (50%–63%) and moderate intensity (64%–76%).[14] Using the mean age of golfer in UK (63 year[45]), the mean range for moderate intensity is 101–119 bpm—the large majority of data fall into this category.

There were varied results in oxygen uptake. In terms of $V_{O_2}max$, studies classified golf as light (37%–45% $V_{O_2}max$), moderate (46%–63% $V_{O_2}max$) and vigorous (64%–90% $V_{O_2}max$).[44] Many studies were close to, but did not reach, the moderate intensity threshold of $10.5$–$20.7 \, mL \, kg^{-1} \, min^{-1}$ (3–5.9 METs) and would therefore be classified as light activity ($<10.5 \, mL \, kg^{-1} \, min^{-1}$).

All included studies, on average, attained the often cited 10 000 steps[44] during an 18-hole round and, according to Tudor-Locke et al,[46] would be classed as moderate-to-vigorous PA. Distance walked is highly variable depending on the course; values range from 6.4 to 11.3 km for an 18-hole round and 4.4–5.3 km for a 9-hole round. In relation to strength, flexibility and balance, greater strength and range of motion were found in those with higher proficiency.[39] It is unclear whether this is due to increased volume of play, additional strength/flexibility work or whether these characteristics are likely to lead to a lower handicap. Furthermore, there appears to be better balance control in golfers. The complex motion while swinging a club and/or walking on uneven grounds during golf play may lead to improved stability; however, this cannot be proven due to the methods employed in this study.

Evidence suggests that use of an electric golf cart significantly reduces PA attained in terms of EE, HR and distance covered. Males expend more energy and walk further distances than females. However, it is likely that this difference is due to greater body mass and longer course length played by males. When course length is accounted for, women walk significantly longer.[33] Skill level does not appear to affect PA accrued, with the possible exception of strength, balance and flexibility. The evidence is unclear whether course profile and age affect PA accrued.

This study is, to our knowledge, the first systematically conducted review to focus exclusively on golf and PA. It provides a general overview of PA accrued while playing golf. A rapid review was conducted due to time constraints. Rapid reviews make use of streamlined methods and, due to this, are not subject to the same rigour as systematic reviews. For some outcomes, there was little available evidence. Furthermore, the sample sizes of included studies were generally small and ranged from 6 to 257 (median 22).

In agreement with the recent scoping review[10] and the Compendium of Physical Activities,[11] golf can provide moderate intensity PA. Exercise intensity varies during the game itself. For certain populations, it may be primarily a low-intensity PA. Shortfalls in intensity, however, are compensated for by the length of the game. Therefore, golf is a viable sport by which to achieve the PA recommendations.[1] Golfers may find it difficult to play enough during a week in order to reach PA recommendations and may wish to supplement golf with another PA. Clinicians and policy-makers can be encouraged to suggest golf as a form of PA in order to meet recommended levels and attain health benefits.

Further research is warranted to investigate whether strength and flexibility is accrued while playing golf as well as research examining the effect of modifiers such as age, course profile, disease characteristics and carrying or pulling clubs, on PA attained.

## CONCLUSION

This rapid review identified 19 articles that examined golf and PA. Golf is primarily a moderate intensity PA, but may be low intensity or even high intensity depending on the population and various modifiers present. If able, golfers should walk the course, rather than ride a golf cart to maximise health benefits. Course profile, skill level and age may affect the amount of PA accrued, and further research is required.

**Contributors** JL: Lead researcher. Performed study selection, data extraction, quality assessment and narrative synthesis. ADM: Primary supervisor. Provided advice on methods including study selection criteria, data extraction and presentation of data. Performed phase one of the search strategy (previously published). EJ: Independently reviewed 10% of paper for study selection, data extraction and quality assessment. DA: Senior researcher. Performed phase two of search strategy and phase two study selection by title and abstract. Provided advice on methods including study selection criteria, data extraction and quality assessment.

**Funding** This work was supported by the Medical Research Council (MRC; MR/K023209/1).

**Competing interests** Although not for this project, ADM has previously received funding to complete research from the World Golf Foundation. The World Golf Foundation committed to publishing whether results were positive, negative or equivocal and had no influence on the conduct of this or previous research.

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
