## [Reviewer comments · BMJ Open]

ARTICLE DETAILS

TITLE (PROVISIONAL)	A rapid review to identify physical activity accrued whilst playing golf
AUTHORS	Luscombe, Jack; Murray, Andrew; Jenkins, Evan; Archibald, Daryll

VERSION 1 – REVIEW

REVIEWER	Maria Stokes, Professor of Musculoskeletal Rehabilitation Faculty of Health Sciences, University of Southampton, United Kingdom One of the authors is known to me (AM) but the field of golf and health is relatively small. I therefore do not consider that there are any competing interests to declare.
REVIEW RETURNED	28-Aug-2017

GENERAL COMMENTS	This paper describes a rapid review of the literature to determine physical activity involved in playing golf and modifiers that affect the activity undertaken. The authors have justified their study design adequately. However, inclusion of 'poor' papers is debatable (see specific points 5 and 13 below). The paper is presented clearly, generally well written and describes literature demonstrating evidence that playing golf can be encouraged to achieve physical activity recommendations. SPECIFIC POINTS (page number refers to number at bottom of each page) 1. Moderate to vigorous physical activity (MVPA) is also used to measure physical activity, so some papers might have been missed. Is there a reason that MVPA was not included in the search?2. Pg 3, line 34: 10% of the papers rather than paper3. Pg 4, line 10: data were, rather than was.4. Pg 4, line 12: should this read, 'further random sample of 10%' ?5. Pg 4, lines 33-34: Quality assessment. It is unclear how inclusion of 'poor' studies could give interesting insights which, in the case of the present study, did not occur. The only value their inclusion appeared to bring to the study was to increase the sample, which was limited in size. Two 'poor' papers concurred with better quality papers, so either way, their inclusion or exclusion would not affect the study conclusions. The 'poor' paper which showed no significant differences is difficult to justify, particularly without any discussion of how the quality of the study could have affected the negative results, e.g. the study could have been under powered, therefore not making it possible to draw a definitive conclusion and hence justifying its exclusion from the study. The problem with including poor studies, is it increases their citation rates for the wrong reasons.
---

6. Pg 5, Figure 1. The flow diagram is confusing at the point of full text articles assessed. After seeing that 168 of 170 papers were excluded, the reader would usually see n=2. Instead, the next box shows n=51, as 49 papers have been added from another box on the top right side of the diagram. A step is therefore missing. To avoid the reader having to work out what's happened, simply adding n=2 next to the arrow between the boxes for 'Records screened' and 'Full texts articles assessed' would be very helpful.

7. Pg 6, line 11: it is usual to avoid starting sentences with a number and to write it as a word. This comment applies to other areas of the manuscript e.g. Pg 9, line 41, so please check throughout.

8. Tables. It would be helpful to know the sample sizes and ages of participant groups when looking at the tables. Since these are given in Appendix 7, it would help to refer the reader to the appendix for this information.

9. Pg 8. lines 11-12: there is a change of tense between 'found' and 'notes' when referring to previous studies. Please decide which throughout e manuscript and be consistent.

10. Pg 12, line 48: the strength between the two sides of the body cannot decrease but is greater on one side than the other.

11. Pg 13, lines 47-48: states: 'The exclusion of these studies is therefore unlikely to have significantly affected the results.' This reads as though these two papers were actually excluded. Suggest rephrasing to e.g:

'The exclusion of these studies would therefore have been unlikely to affect the results significantly'.

12. Pg 14, line 30: 'there was a significant decrease in right and left handgrip strength in golfers than non-golfers'. Suggest rephrase to, e.g:

'handgrip strength was significantly greater on the left than on the right side in golfers compared to non-golfers'. Or

'handgrip strength was significantly lower on the right than on the left side

This point also applies on line 55, in relation to differences in body weight between males and females i.e. greater in males, rather than increased.

13. Pg 14, lines 16-17: it is difficult to see why the fact that the third 'poor' study showed 'divergent' results from other studies justified its inclusion or how it contributed insights. One of the few interesting results that demonstrated a difference was greater handgrip on the left side. However, this was not novel and reasons for this, related to the golf awing technique, had been explained in the original study by Barnes & Adams in 2007 (Barnes J, Adams J. Differences in dominant and non-dominant handgrip strength of male golf professionals measured using the Jamar Dynamometer. British Journal of Hand Therapy. 2007;12(4):112-116).

14. Discussion section - a comment on the sample sizes of the studies included would be useful.

15. Pg 14, lines 46-49: the observation that golfers had better balance than ono-golfers may be a training effect of playing golf but this cannot be concluded from a cross-sectional study and requires a prospective study of novice golfers to determine cause and effect.

	16. Pf 15, line 39. 'If able, golfers should walk the course, either pulling or carrying clubs,'. Caution is needed in recommending pulling or carrying clubs, as the implications of these actions have not been considered in the paper. Pulling a golf trolley differs biomechanically from pushing a trolley and anecdotally, those with shoulder pain and back pain find it easier and less painful to push than pull a trolley, although this requires investigation. More importantly, the effect of carrying clubs on the spine, particularly in the young, whose spines are still developing, needs to be considered. Also, carrying clubs centrally on the back may be preferable for the shoulders, neck and back than unilateral loading by carrying on one side. Without consideration of the potential risks of different ways of transporting clubs, it may be appropriate to simply state that it is advisable that golfers walk the course rather than ride in a golf cart to maximise health benefits.
--	---

REVIEWER	Conor Gissane School of Sport Health and Applied Science St Mary's University Twickenham Middlesex TW1 4SX
REVIEW RETURNED	31-Aug-2017

GENERAL COMMENTS	The paper seeks to produce a systematic review of physical activity accrued whilst golfing, and to assess potential moderating factors. I appreciate that it is a rapid review, but attention to detail is still important. I appreciate the volume of information that the authors are trying to present. But, on several occasions detail was lacking, and some points were made too casually. I also kept thinking that a meta-analysis would help, but that was not an aim of the paper. Start sentences with capital letters not numbers. Starting with numbers is only acceptable in an abstract. Page 3 Introduction 1st sentence You wrote “Physical activity guidelines generally recommend, for adults, at least 150 minutes of moderate intensity activity, or 75 minutes of vigorous physical activity per week, or a combination of the two.” That is a lot of clauses for the opening sentence. If you want the reader to take it all in, please consider revising it. Can I suggest that this sentence “Moderate intensity physical activity is known to provide longevity, physical and mental health benefits”, currently on lines 10-11 goes first? Page 3 Line 19 You wrote
--

"In contrast to the majority of sports,"

Please change to:

"In contrast to most sports,"

Page 4 line 3-4

You wrote:

"...only one reviewer assessed all papers for selection, data extraction and quality assessment compared to a full systematic review."

I did consult the manual, your reference 10. It does not say "one reviewer", but uses the term "reviewers". True, your earlier study did use the same strategy as this study, but that is the only reference I can find for prior use.

Page 4 line 25-6

I like the fact that you have included your searches in an appendix.

Page 6 lines 6 -11

There are two sentences in this paragraph that start with numbers. Sentences should start with a capital letter.

Page 6 figure 1

This is a little confusing. The diagonal line from the box "records identified through phase 1 search" is certainly not standard. I know in the text you make reference to appendix 4, but could you put "(see appendix 4)" in the phase one search box? That way it is noted that this search was supplemental.

Page 7 lines 5 - 9

Again you have sentences starting with numbers.

Page 7 line 11

As above

Page 7 lines 30-34

As above

Page 7 lines 43-44

Can you expand this paragraph? At the moment it looks as though there is very little to say and you have to question if it needs to be there.

Page 8 table 1

None of the listed studies in this table has the number of subjects included. It is an important to note this, or are they a series of case studies?

Page 9 table 2

Please include sample sizes.

Page 9

Lines 41-2

More sentences beginning with numbers.

Page 11 table 3

Sample sizes would help

Page 12 line 5

Yet another sentence beginning with a number

Page 12 line 48

Begin sentence with capital letters.

Page 13 line 3

Begin sentence with capital letters.

Page 13 lines 17-19

You wrote:

“Tangen et al. 33 found that, when course length is accounted for, women (2.13x course length) walked significantly longer than men (1.98x course length).”

What exactly do 2.13 and 1.98 refer to? What are the units of measurement?

Page 13 Table 5

Sample sizes would help.

Page 15 line 8

You wrote

“age of golfers in UK (63yrs...)”

Units of measurement are never plural.

Page 15 lines 52 - 53

You wrote:

“Males expend more energy and walk further distances than females.”

This appears to contradict your point made on page 13 lines 17-19, where you said women walked further.

	Page 16 line 9 “Rapid reviews make use streamlined methods.” I think the “of” is missing.
--	--

VERSION 1 – AUTHOR RESPONSE

Reviewer: 1

Reviewer Name: Maria Stokes, Professor of Musculoskeletal Rehabilitation

Institution and Country: Faculty of Health Sciences, University of Southampton, United Kingdom

Competing Interests: One of the authors is known to me (AM) but the field of golf and health is relatively small. I therefore do not consider that there are any competing interests to declare.

GENERAL COMMENTS

This paper describes a rapid review of the literature to determine physical activity involved in playing golf and modifiers that affect the activity undertaken. The authors have justified their study design adequately. However, inclusion of 'poor' papers is debatable (see specific points 5 and 13 below). The paper is presented clearly, generally well written and describes literature demonstrating evidence that playing golf can be encouraged to achieve physical activity recommendations.

Response: We thank the reviewer for their comments and insights. Please see below for our responses to specific points.

SPECIFIC POINTS (page number refers to number at bottom of each page)

1. Moderate to vigorous physical activity (MVPA) is also used to measure physical activity, so some papers might have been missed. Is there a reason that MVPA was not included in the search?

Response: We thank the reviewer for this point. The search strategy for the review was conducted in regular consultation with a highly experienced research librarian. Inclusivity was maximised by having “golf” as the only search term for the health focused databases, to maximise inclusivity. Although the point made is a fair one, we are confident that relevant studies have been identified by the employed search strategy.

2. Pg 3, line 34: 10% of the papers rather than paper

Response: Amended.

3. Pg 4, line 10: data were, rather than was.

Response: Amended.

4. Pg 4, line 12: should this read, 'further random sample of 10%' ?

Response: Yes, thank you, changed.

5. Pg 4, lines 33-34: Quality assessment. It is unclear how inclusion of 'poor' studies could give interesting insights which, in the case of the present study, did not occur. The only value their inclusion appeared to bring to the study was to increase the sample, which was limited in size. Two 'poor' papers concurred with better quality papers, so either way, their inclusion or exclusion would not

affect the study conclusions. The 'poor' paper which showed no significant differences is difficult to justify, particularly without any discussion of how the quality of the study could have affected the negative results, e.g. the study could have been under powered, therefore not making it possible to draw a definitive conclusion and hence justifying its exclusion from the study. The problem with including poor studies, is it increases their citation rates for the wrong reasons.

Response: We thank the reviewer for their comments on the inclusion of 'poor' studies. In hindsight, we agree that the inclusion of these studies is not beneficial to the review and, as suggested, has the detrimental effect of increasing their citation rates.

The 3 'Poor' papers that were originally included (Getchell 1965, Moy 2006, Murase 1989) have now been excluded. The manuscript and Appendices 5, 6 and 7 have been adjusted throughout to reflect this.

6. Pg 5, Figure 1. The flow diagram is confusing at the point of full text articles assessed. After seeing that 168 of 170 papers were excluded, the reader would usually see n=2. Instead, the next box shows n=51, as 49 papers have been added from another box on the top right side of the diagram. A step is therefore missing. To avoid the reader having to work out what's happened, simply adding n=2 next to the arrow between the boxes for 'Records screened' and 'Full texts articles assessed' would be very helpful.

Response: Thank you for the suggestion, n=2 has been added next to the correct arrow.

7. Pg 6, line 11: it is usual to avoid starting sentences with a number and to write it as a word. This comment applies to other areas of the manuscript e.g. Pg 9, line 41, so please check throughout.

Response: Thank you, this has been corrected throughout the document.

8. Tables. It would be helpful to know the sample sizes and ages of participant groups when looking at the tables. Since these are given in Appendix 7, it would help to refer the reader to the appendix for this information.

Response: We agree this information would be helpful. The caption "Please refer to Appendix 6 for characteristics of the above studies." has been added to all tables to direct readers to this information.

9. Pg 8. lines 11-12: there is a change of tense between 'found' and 'notes' when referring to previous studies. Please decide which throughout the manuscript and be consistent.

Response: Amended to "Zunzer et al.³² noted no significant difference in METs between sexes; whereas Tangen et al.³³ found an almost significant difference..."

The tense has been amended throughout the manuscript to provide consistency.

10. Pg 12, line 48: the strength between the two sides of the body cannot decrease but is greater on one side than the other.

Response: This line has now been removed as studies rated 'Poor' have now been excluded from the review (see above).

11. Pg 13, lines 47-48: states: 'The exclusion of these studies is therefore unlikely to have significantly affected the results.' This reads as though these two papers were actually excluded. Suggest rephrasing to e.g:

'The exclusion of these studies would therefore have been unlikely to affect the results significantly'.

Response: This line has now been removed as studies rated 'Poor' have now been excluded from the review (see above).

12. Pg 14, line 30: 'there was a significant decrease in right and left handgrip strength in golfers than non-golfers'. Suggest rephrase to, e.g:

'handgrip strength was significantly greater on the left than on the right side in golfers compared to non-golfers'. Or

'handgrip strength was significantly lower on the right than on the left side

This point also applies on line 55, in relation to differences in body weight between males and females i.e. greater in males, rather than increased.

Response: Both of these lines have now been removed as studies rated 'Poor' have now been excluded from the review (see above).

13. Pg 14, lines 16-17: it is difficult to see why the fact that the third 'poor' study showed 'divergent' results from other studies justified its inclusion or how it contributed insights. One of the few interesting results that demonstrated a difference was greater handgrip on the left side. However, this was not novel and reasons for this, related to the golf swing technique, had been explained in the original study by Barnes & Adams in 2007 (Barnes J, Adams J. Differences in dominant and non-dominant handgrip strength of male golf professionals measured using the Jamar Dynamometer. British Journal of Hand Therapy. 2007;12(4):112-116).

Response: We thank the reviewer for this helpful insight. All 'Poor' studies have now been excluded (see above).

14. Discussion section - a comment on the sample sizes of the studies included would be useful.

Response: We thank the reviewer for this suggestion and agree it would be useful. The comment "Furthermore, the sample sizes of included studies was generally small and ranged from 6 – 257 (median 22)" has been added.

15. Pg 14, lines 46-49: the observation that golfers had better balance than non-golfers may be a training effect of playing golf but this cannot be concluded from a cross-sectional study and requires a prospective study of novice golfers to determine cause and effect.

Response: Many thanks for the comment. We were not attempting to conclude this, but more make a suggestion for this observation. However, the phrasing may be too definitive. This has been amended to provide more clarity: "Furthermore, there appears to be better balance control in golfers. The complex motion while swinging a club and/or walking on uneven grounds during golf play may lead to improved stability, however this cannot be proven due to the methods employed in this study."

16. Pf 15, line 39. 'If able, golfers should walk the course, either pulling or carrying clubs,

Caution is needed in recommending pulling or carrying clubs, as the implications of these actions have not been considered in the paper. Pulling a golf trolley differs biomechanically from pushing a trolley and anecdotally, those with shoulder pain and back pain find it easier and less painful to push than pull a trolley, although this requires investigation.

More importantly, the effect of carrying clubs on the spine, particularly in the young, whose spines are still developing, needs to be considered. Also, carrying clubs centrally on the back may be preferable for the shoulders, neck and back than unilateral loading by carrying on one side. Without

consideration of the potential risks of different ways of transporting clubs, it may be appropriate to simply state that it is advisable that golfers walk the course rather than ride in a golf cart to maximise health benefits.

Response: Thanks for your insight into this. This has been amended to "If able, golfers should walk the course, rather than ride a golf cart to maximise health benefits."

Reviewer: 2

Reviewer Name: Conor Gissane

Institution and Country: School of Sport Health and Applied Science, St Mary's University, Twickenham, UK

Competing Interests: None

The paper seeks to produce a systematic review of physical activity accrued whilst golfing, and to assess potential moderating factors. I appreciate that it is a rapid review, but attention to detail is still important. I appreciate the volume of information that the authors are trying to present. But, on several occasions detail was lacking, and some points were made too casually. I also kept thinking that a meta-analysis would help, but that was not an aim of the paper.

Response: We thank the reviewer for their comments. After discussion, we felt a meta-analysis was not feasible due to the heterogeneity of results and that narrative synthesis would allow for better presentation of data. Please see below for our responses to specific points.

Comment: Start sentences with capital letters not numbers. Starting with numbers is only acceptable in an abstract.

Response: Thank you. This has been amended throughout the document.

Page 3 Introduction 1st sentence

You wrote

"Physical activity guidelines generally recommend, for adults, at least 150 minutes of moderate intensity activity, or 75 minutes of vigorous physical activity per week, or a combination of the two."

That is a lot of clauses for the opening sentence. If you want the reader to take it all in, please consider revising it.

Can I suggest that this sentence

"Moderate intensity physical activity is known to provide longevity, physical and mental health benefits", currently on lines 10-11 goes first?

Response: Thank you for this point, the suggested sentence has been made the opening sentence.

Page 3 Line 19

You wrote

"In contrast to the majority of sports,"

Please change to:

"In contrast to most sports,"

Response: Amended.

Page 4 line 3-4

You wrote:

“...only one reviewer assessed all papers for selection, data extraction and quality assessment compared to a full systematic review.”

I did consult the manual, your reference 10. It does not say “one reviewer”, but uses the term “reviewers”. True, your earlier study did use the same strategy as this study, but that is the only reference I can find for prior use.

Response: The quote above describes the concessions made due to this being a rapid review. These concessions have previously been described in the literature. Reference 10 is the previous scoping review on golf & health. We’ve added a line following this and referenced a scoping review of rapid review methods detailing concessions described in other rapid reviews: “Similar concessions have been described in the literature²².”

Page 4 line 25-6

I like the fact that you have included your searches in an appendix.

Response: Thank you.

Page 6 lines 6 -11

There are two sentences in this paragraph that start with numbers. Sentences should start with a capital letter.

Response: We agree and have amended.

Page 6 figure 1

This is a little confusing. The diagonal line from the box “records identified through phase 1 search” is certainly not standard. I know in the text you make reference to appendix 4, but could you put “(see appendix 4)” in the phase one search box? That way it is noted that this search was supplemental.

Response: We agree this would be helpful, “see Appendix 4” has been added to the phase one search box. Thank you.

Page 7 lines 5 - 9

Again you have sentences starting with numbers.

Response: Amended.

Page 7 line 11

As above

Response: Amended.

Page 7 lines 30-34

As above

Response: Amended.

Page 7 lines 43-44

Can you expand this paragraph? At the moment it looks as though there is very little to say and you have to question if it needs to be there.

Response: We thank the reviewer for this well-made point, and have expanded to read:
"Two studies found no significant difference in energy expenditure in relation to skill level, despite less skilled players taking a larger number of shots in total, and on average being less able to advance the ball accurately".

Page 8 table 1

None of the listed studies in this table has the number of subjects included. It is an important to note this, or are they a series of case studies?

Response: We agree that it is important to include sample sizes in the tables. However, we are wary of adding further columns to tables that are already rather large. In agreement with the other reviewer of this paper, the caption "Please refer to Appendix 6 for characteristics of the above studies." has been added to all tables to direct readers to this information.

Page 9 table 2

Please include sample sizes.

Response: See above.

Page 9

Lines 41-2

More sentences beginning with numbers.

Response: Amended.

Page 11 table 3

Sample sizes would help

Response: See above.

Page 12 line5

Yet another sentence beginning with a number

Response: Amended.

Page 12 line 48

Begin sentence with capital letters.

Response: Amended.

Page 13 line 3

Begin sentence with capital letters.

Response: Amended.

Page 13 lines 17-19

You wrote:

"Tangen et al.³³ found that, when course length is accounted for, women (2.13x course length) walked significantly longer than men (1.98x course length)."

What exactly do 2.13 and 1.98 refer to? What are the units of measurement?

Response: In this instance, the course length is the unit of measurement. 2.13 and 1.98 refer to the distance relative to the course length that was walked e.g if the course length was 10km, women walk 21.3km and men 19.8km. We have improved the presentation of this by stating “2.13 times the course length” and “1.98 times the course length”.

Page 13 Table 5

Sample sizes would help.

Response: See above.

Page 15 line 8

You wrote

“age of golfers in UK (63yrs...”

Units of measurement are never plural.

Response: Thank you, amended to “age of golfer in UK...”

Page 15 lines 52 - 53

You wrote:

“Males expend more energy and walk further distances than females.”

This appears to contradict your point made on page 13 lines 17-19, where you said women walked further.

Response: The point made earlier (page 13 lines 17-19) says that “when course length is accounted for, women (2.13 times the course length) walked significantly longer than men (1.98 times the course length). The paragraph in question goes on to explain this “When course length is accounted for, women walk significantly longer”. The point being made here is that although the absolute distance walked is further in men, the distance walked relative to the course length is further in women.

Page 16 line 9

“Rapid reviews make use streamlined methods.” I think the “of” is missing.

Response: Amended to “Rapid reviews make use of streamlined..”

VERSION 2 – REVIEW

REVIEWER	Maria Stokes, Professor of Musculoskeletal Rehabilitation Faculty of Health Sciences, University of Southampton, United Kingdom One of the authors is known to me (AM) but the field of golf and health is relatively small. I therefore do not consider that there are any competing interests to declare.
REVIEW RETURNED	21-Oct-2017

GENERAL COMMENTS	The authors appear to have address the comments raised by the reviewers and the manuscript has been improved.
---

REVIEWER	Conor Gissane St Mary's University Twickeham Middlesex TW7 5DU UK
REVIEW RETURNED	13-Oct-2017

GENERAL COMMENTS	The authors have addressed each the points I raised. There is one point of miscommunication outlined below. I will take responsibility for that. Well done to the authors Page 14 line 21 to 22 Thank you for the changes, but changing “golfers” to “golfer” was not what I meant. When I said units of measurement are never plural, I was referring to “yrs” and that it should be changed to “yr”. I will take responsibility for the unclear communication. Can you change it to, “Using the mean age of a golfer in the UK (63yr),...”
---